# SRAttack: Super-Resolution as an Attack on Invisible Image Watermarking

## Abstract

Invisible image watermarking is an important technology for copyright protection, enabling the sharing and spread of invisible images. The study of watermark attacks motivates the development of more robust invisible image watermarking. An ideal attack should effectively erase the watermark, preserve the visual quality of the image, and be simple and accessible. Although existing methods can partially destroy watermarks, they often fail to preserve satisfactory visual quality of image. In this paper, we propose the Super-Resolution Attack (SRAttack), a simple attack framework that leverages super-resolution (SR) to overcome these limitations. The proposed attack method first pre-degrade the watermarked image, then use a SR model to reconstruct a high-quality, watermark-free version, and finally down-sample it back to original dimensions of cover image. SRAttack operates in a black-box, training-free manner, and can be readily instantiated. Extensive experiments demonstrate that SRAttack not only effectively destroys watermarks embedded by various advanced watermark algorithms, but also improves visual quality in contrast to existing attack methods. Our research findings reveal a new perspective on invisible image watermarking research, indicating the need to consider attacks aimed at improving visual quality rather than just reducing it. Our code will be available soon.

## 1 Introduction

As a crucial method for copyright protection, invisible image watermarking technology embeds information into images in a nearly imperceptible manner, thereby simultaneously achieving ownership authentication and content tracking (Hosny et al., 2024). The core challenge is to develop a watermark that is both robust against various attacks and imperceptible (Rabbani et al., 2024), while maintaining the visual quality of the image. The study of watermark attacks can reveal the limitations of current invisible image watermarking and inspire the development of more robust invisible image watermarking.

In practice, attackers often aim to easily remove watermarks while preserving the image's usability, meaning that they prefer methods that maintain high visual quality of image. Existing attack approaches exhibiting distinct advantages and limitations. White-box (Nam et al., 2025) and gray-box attacks (Saberi et al., 2024) demonstrate effectiveness when attackers have access privileges or possess knowledge of the watermarking scheme, yet their practical utility remains constrained by the difficulty of acquiring such privileged information. More accessible black-box attacks, including geometric transformations (Zhou et al., 2023) and JPEG compression (Jia et al., 2021), operate without requiring embedding details. However, these watermark attack strategies have a critical limitation: they primarily work by reducing the image's visual quality (Rabbani et al., 2024). The emerging reconstruction-based approaches (Zhao et al., 2024), which employ generative models such as VAEs or diffusion models, can successfully remove watermarks while maintaining visual fidelity, yet fail to enhance image quality. The limitations of these methods have left watermarking systems vulnerable to a new type of threat—attacks that use advanced image reconstruction techniques to erase watermarks without degrading quality.

Super-resolution reconstruction is a highly effective method for enhancing the visual quality of images (Wang et al., 2020; Su et al., 2025). Our central insight stems from the fundamental working principles of both invisible image watermarking and Super-Resolution (SR) models. As shown in

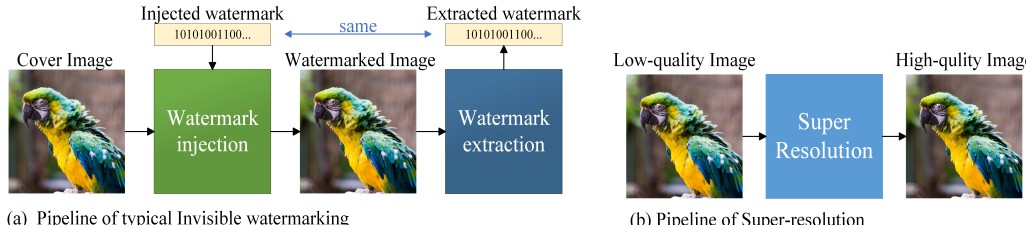

(a) Pipeline of typical Invisible watermarking

(b) Pipeline of Super-resolution

Figure 1: Pipeline of invisible image watermarking and super-resolution.

Figure 1 (a), a typical invisible watermarking pipeline embeds binary information into an image in a way that is imperceptible to the human eye, which a watermark extractor can subsequently recover. To maintain invisibility, watermarks are often embedded in image details (Luo et al., 2023b), including edges (Jiang et al., 2024), complex textures (Huang et al., 2023), or even smooth regions(Tancik et al., 2020). As shown in Figure 1 (b), an SR pipeline is designed to reconstruct a high-fidelity image from a low-quality input, removing artifacts (Wang et al., 2022a) and restoring texture detail (Fan et al., 2025) from blurred regions. Since watermarks are similar to artifacts or noise, the SR model can naturally remove them during image restoration.

Based on this discovery, we propose a universal and effective attack framework called Super-Resolution Attack (SRAttack). SRAttack begins by degrading the watermarked image to simulate a low-quality input for the Super-Resolution (SR) model. This degraded image is then processed by the SR model, which can effectively remove the watermark during its reconstruction process. Finally, a downscaling operation is applied to the output image to restore it to its original dimensions. This framework works in a black-box setting and does not require algorithm details of invisible image watermarking, or additional watermarked examples. It is also training-free and easy to implement with public tools.

The main contributions of this paper are as follows:

- This study is the first to systematically analyze and evaluate with super-resolution for watermark removal in invisible image watermarking. Our analysis reveals a critical vulnerability in the robustness of existing watermarking methods when subjected to super-resolution-based attacks.

- We propose a universal and efficient attack framework termed SRAttack, which primarily comprises three key steps: Pre-degradation, Super-Resolution Reconstruction, and Downscaling. This framework operates in a black-box manner, follows a training-free paradigm, and can be readily instantiated.

- Comprehensive experiments demonstrate the effectiveness of SRAttack. Our results on benchmark datasets show that SRAttack successfully attacks the state-of-the-art watermark detection techniques. Compared to existing attack methods, it effectively destroys watermarks while maintaining superior visual quality of images.

## 2 RELATED WORK

This section reviews prior work in invisible image watermarking, watermark attacks, and super-resolution, which are the three core areas of our research.

### 2.1 INVISIBLE IMAGE WATERMARKING

Invisible watermarking aims to embed imperceptible information into images for purposes like copyright protection. The primary challenge lies in balancing imperceptibility, robustness, and capacity (Wan et al., 2022). Early methods often manipulated frequency domain coefficients (e.g., DCT, DWT), but they typically struggled with robustness against various image distortions (Hsu & Wu, 1999; Navas et al., 2008).

Deep learning has significantly advanced the field, particularly through end-to-end models. The framework introduced by Zhu et al. (2018), which uses an encoder-decoder architecture trained against a set of simulated attacks, became a foundational paradigm. Subsequent research has built upon this framework. For example, Ahmadi et al. (2020) incorporated a residual design, and Jia et al. (2021) enhanced robustness specifically against JPEG compression. This paradigm was also extended to video watermarking in models (Zhang et al., 2019; Luo et al., 2023a). Tancik et al. (2020) demonstrated remarkable robustness against real-world print-and-scan operations.

Another prominent line of research leverages Invertible Neural Networks (INNs) to achieve high imperceptibility and lossless data embedding. To improve the robustness of INN-based methods, Yang et al. (2024) introduced enhancement modules and a multi-step training strategy to defend noise and compression attacks. Ma et al. (2022) proposed a hybrid approach, combining invertible networks with non-invertible, noise-resistant modules to enhance overall robustness. Similarly, Kou et al. (2025) was designed for robustness against noise, employing a Deform-Attention based INN and a learnable wavelet network to embed the watermark in less perceptible high-frequency regions. However, the reliance of these advanced methods on high-frequency details makes them potentially vulnerable to the super-restoration-based attack.

## 2.2 IMAGE SUPER-RESOLUTION

Image Super-Resolution (ISR) is a classic computer vision task that aims to reconstruct a high-resolution (HR) image from its low-resolution (LR) counterpart (Nasrollahi & Moeslund, 2014). While early SR methods were based on interpolation or traditional reconstruction, the field is now dominated by deep learning approaches that learn a direct mapping from LR to HR images, yielding significantly more realistic results (Wang et al., 2020; Lepcha et al., 2023). GAN-based models, such as ESRGAN (Wang et al., 2018) and its successor Real-ESRGAN (Wang et al., 2021b), have become widely applied for their ability to generate perceptually convincing textures.

Recently, significant attention has been drawn to diffusion models. These models generate images through a denoising process and have shown a remarkable capability for rendering realistic textural details. However, this process is often computationally intensive. For instance, SeeSR (Wu et al., 2024b) typically requires 50 diffusion steps to generate an image. To enhance efficiency, Wu et al. (2024a) using the low-resolution (LR) image as the starting point for the diffusion process instead of random noise, thereby achieving high-quality SR in just a single step and dramatically speeding up inference. Building on this, AdcSR (Chen et al., 2025) further distills the model to create a more streamlined architecture while preserving its strong generative power. Leveraging the generative capabilities of Diffusion models for image super-resolution reconstruction is a highly effective approach. During the reconstruction process, efficiency stands as a significant factor worthy of consideration, as lower computational requirements greatly facilitate the widespread adoption of the method.

## 2.3 WATERMARKING ATTACKS

The objective of a watermark attack is to remove the watermark without compromising the visual quality of the image. These attacks can be classified based on the attacker's knowledge and access level. For instance, in a white-box setting, some attacks presuppose that the watermarking algorithm is fully accessible (Nam et al., 2025), or the watermark detector is real-time accessible (Jiang et al., 2023). Gray-box attacks operate with partial knowledge, which involve obtaining watermarked/original image pairs from the target model (Saberi et al., 2024), or leveraging a surrogate model with a similar structure (Lukas et al., 2023). However, the practical applicability of these methods is limited, as detailed information or access privileges are typically difficult for an attacker to obtain.

In contrast, black-box attacks, such as geometric transformations, noise addition, brightness and contrast adjustments, blurring, and JPEG compression are more accessible as they ignore how the watermark is embedded. Some researchers utilize advanced learning-based Variational Autoencoder (VAE) compression techniques (Ballé et al., 2018; Cheng et al., 2020) to compress images for attacking watermarks. However, such methods often degrade the visual quality of image. Recently, reconstruction attacks have emerged (Saberi et al., 2024; An et al., 2024). For example, Lukas & Kerschbaum (2023) explored the removal of deepfake image generator watermarks using pre-trained

Figure 2: The conceptual diagram of the Super-Resolution Attack (SRAttack) framework. The attack pipeline consists of an optional pre-degradation step, a core super-resolution reconstruction step defined by a powerful image prior, and a final downscaling step.

latent diffusion models. Zhao et al. (2024) adds noise to the latent representation of a watermarked image and then leverages the reconstruction capability of a VAE model or the denoising process of a diffusion model to generate a watermark-free image. However, these reconstruction attacks are effective at preserving visual quality, but do not enhance it. Since our objective of attacking watermarks is to enable the use of images without watermarks, the quality of the attacked images is of great importance. Therefore, it's need to study new attack methods that not only removes watermarks but also improves image quality.

## 3 METHOD

In this section, we first outline our Problem Definition and Threat Model. Subsequently, we present the detailed principles of the method and describe its instantiation.

### 3.1 PROBLEM DEFINITION AND THREAT MODEL

**Problem Definition.** Watermark attacks aim to remove invisible watermarks from watermarked images while preserving their usability. These watermarked images are produced by watermarking algorithms, which represent the threat model addressed by our proposed method.

Formally, an attack is defined as applying a transformation to the watermarked image $I_{wm}$, yielding an attacked image $I_a$. The attack is deemed successful if it induces the decoder to recover a message $m'$ that shows substantial divergence from the original embedded message $m$.

**Threat Model.** Invisible image watermarking is a technique for embedding a secret message into a cover image in a visually imperceptible manner. Formally, a watermarking system (threat model) consists of an encoder $\mathcal{E}$ and a decoder $\mathcal{D}$. The encoder embeds a binary message $m \in \{0,1\}^L$ into a cover image $I_{cover} \in \mathbb{R}^{H \times W \times 3}$, generating a watermarked image $I_{wm} = \mathcal{E}(I_{cover}, m)$. The decoder's task is to recover the message $m' = \mathcal{D}(I_a)$ from a potentially attacked image $I_a$.

### 3.2 SUPER-RESOLUTION ATTACK

We propose a watermark attack framework, Super-Resolution Attack (SRAttack), which leverages the inherent properties of Super-Resolution (SR) models. The core idea is to leverage the super-resolution process to both destroy watermarks and improve the visual quality of watermarked images. Watermarks are embedded into the texture and smooth areas of an image as imperceptible disturbances. When super-resolution (SR) techniques are applied on watermarked images, the SR model is capable of proactively suppressing artifacts that embed watermark information, while concurrently synthesizing novel and genuine texture details to improve overall image quality. This com-

bined process of artifact removal and detail regeneration effectively interferes with the watermark decoding mechanism.

The Super-Resolution Attack (SRAttack) comprises three key steps: Pre-degradation, Super-Resolution as a Watermark Attacker, and Downscaling. Pre-degradation intentionally introduces degradation factors to create a low-quality input image. Super-Resolution as a Watermark Attacker employs advanced algorithms to reconstruct a high-quality, watermark-free image, capitalizing on super-resolution models' ability to restore texture details while suppressing watermark-containing artifacts. Downscaling operation adjusts the reconstructed high-resolution image back to its original size for subsequent usability. These three steps are effectively integrated, with the specific implementation process detailed as follows:

$$I_a = \mathcal{D}_\downarrow(\mathcal{SR}(\mathcal{A}_{\text{pre}}(I_{wm}))) \tag{1}$$

where $I_{wm}$ is the watermarked image, $\mathcal{A}_{\text{pre}}$ is a pre-degradation function, $\mathcal{SR}$ is the super-resolution model, and $\mathcal{D}_\downarrow$ is a final downscaling function. Each stage is described below.

**Pre-degradation.** During the training of a Super-Resolution (SR) model, a series of common degradation methods are used to create low-quality inputs. Our framework needs to mimic this degradation process because if a high-quality watermarked image is input directly, the SR model might not apply strong transformations, thus leaving the watermark intact. By first degrading the image, we force the model to actively reconstruct the image content, which facilitates watermark removal. First, we apply the degradation function $\mathcal{A}_{\text{pre}}$ to the watermarked image $I_{wm}$ to simulate a low-quality input, $I_{lr} = \mathcal{A}_{\text{pre}}(I_{wm})$. The degradation function can be composite, an example of which is shown below:

$$I_{lr} = \mathcal{A}_{\text{pre}}(I_{wm}) = \mathcal{D}(I_{wm}; \Theta) = ((I_{wm} \otimes k) \downarrow_d + n)_{\text{JPEG}_q}, \{k, n, d, q\} \subset \Theta \tag{2}$$

where $\mathcal{D}$ is a degradation map $\mathcal{D} : \mathbb{R}^{W \times H \times c} \to \mathbb{R}^{\tilde{W} \times \tilde{H} \times c}$ with $\tilde{W} \leq W$ and $\tilde{H} \leq H$. $\Theta$ contains degradation parameters, including aspects such as blur $k$, noise $n$, downscaling factor $d$, and compression quality $q$.

**Super-Resolution as a Watermark Attacker.** This stage is the core of the attack. SR Model treats the watermark as an undesirable artifact, similar to noise or compression artifacts it was trained to remove. Meanwhile, the SR model reconstructs texture details in the image's blurry regions, which disrupts the decoding of the watermark. The process can be represented as follows:

$$I_{sr} = \mathcal{SR}(I_{lr}) = \mathcal{M}(I_{lr}; \theta), I_{sr} \in \mathbb{R}^{s\tilde{W} \times s\tilde{H} \times c} \tag{3}$$

where $\mathcal{M} : \mathbb{R}^{\tilde{W} \times \tilde{H} \times c} \to \mathbb{R}^{s\tilde{W} \times s\tilde{H} \times c}$ is a SR Model, $s$ is the upscaling factor of SR Model, $\theta$ is the parameters of $\mathcal{M}$. The parameters $\theta$ can be optimized as follows:

$$\theta^* = \arg\min \mathcal{L}(\hat{y}, y) + \lambda\phi(\theta) \tag{4}$$

where $\mathcal{L}$ represents the loss function that estimates the reconstruction quality between the predicted high-resolution image $\hat{y}$ and the ground truth high-resolution image $y$ from the training dataset pairs, $\lambda$ is a trade-off parameter, and $\phi(\theta)$ is the regularization term that constrains the model parameters to prevent overfitting and improve generalization. Detailed analysis of the attack mechanism can be found in Appendix A.

**Downscaling.** The output of the SR model, $I_{sr}$, is usually larger than the original image. In order to restore the image to its original size ($H \times W$) for ease of use, a final downscaling step is required. This is achieved by applying a downscaling function $\mathcal{D}_\downarrow$ to produce the final attacked image, $I_a$.

$$I_a = \mathcal{D}\downarrow(I_{sr}; H, W), \tag{5}$$

where $\mathcal{D}\downarrow : \mathbb{R}^{s\tilde{W} \times s\tilde{H} \times c} \to \mathbb{R}^{H \times W \times c}$ denotes the downscaling function.

**Attack Instantiation.** The SRAttack framework is highly flexible and modular, and each component can be easily instantiated with common, off-the-shelf, easy methods or tools. In detail, the **pre-degradation** ($\mathcal{A}_{\text{pre}}$) can be implemented using common image degradation method, such as

adding Gaussian noise, applying JPEG compression, or simply downscaling with a basic algorithm, and their combinations. The **super-resolution** ($\mathcal{SR}$) operator can be any pre-trained, state-of-the-art model or off-the-shelf tools. Finally, the **downscaling** ($\mathcal{D}_\downarrow$) can be performed using standard interpolation algorithms like bicubic, bilinear, or Lanczos. This flexibility makes the SRAttack framework general and easy to implement. We have compiled a list of SR methods and tools for instantiation, which is presented in Appendix B.

## 4 EXPERIMENTS

In this section, we conduct experiments to evaluate our proposed Super-Resolution Attacks (SRAttack). These experiments aim to answer the following questions:

**RQ1:** To what extent does SRAttack demonstrate effective watermark removal capabilities compared to existing attack methods?

**RQ2:** How about the visual quality of the images after being attacked by SRAttack?

**RQ3:** Can pre-degradation enhance the attack effectiveness of SRAttack? What distinct effects are achieved when employing different pre-degradation operations?

**RQ4:** How does SRAttack's attack efficacy vary under different hyperparameter configurations? What is the corresponding impact on the visual quality of generated images?

### 4.1 EXPERIMENTAL SETTINGS

**Datasets.** Our experiments are conducted on a diverse set of images to ensure the generality of our findings. We construct our test set by randomly sampling 5,000 images from the MS-COCO (Lin et al., 2014) training set, 5,000 images from the WikiArt dataset, and 500 high-resolution images from the DIV2K (Timofte et al., 2017) dataset. This combination provides a rich variety of natural scenes, artistic styles, and high-quality photographs. For all experiments, we randomly crop a patch from each image and resize it to a standard resolution of $256 \times 256$ pixels to serve as the input for the watermarking systems. Some threat models requiring training, the related training dataset configuration are detailed in Appendix C.1.

**Watermarking Setting.** We evaluate a series of state-of-the-art watermarking algorithms spanning different paradigms: traditional methods (**DwtDct**, **DwtDctSvd** (Navas et al., 2008)), encoder-decoder models (**RivaGAN** (Zhang et al., 2019), **StegaStamp** (Tancik et al., 2020)) and INN-based models (**IWRN** (Kou et al., 2025), **CIN** (Ma et al., 2022)). All images were individually marked with random 32-bit watermarks. Detailed descriptions of each algorithm are provided in Appendix C.2. All evaluations use the publicly available code for each method.

**Baseline Attack Methods.** We conducted a comprehensive comparative evaluation of SRAttack, selecting 11 challenging attack methods as baseline: image-processing attacks, including Gaussian Blur (kernel size 5, std 1), Gaussian Noise (std 0.05), Contrast adjustment (factor 0.1), Brightness adjustment (factor 0.1), Rotation (15°), Resize (0.25×), JPEG compression (q=50), and BM3D denoising (Dabov et al., 2007) (std 0.9); learned compression models using VAE implementations from Bmshj (Ballé et al., 2018) and Cheng (Cheng et al., 2020) (quality 5); a generative regeneration attack based on a diffusion model (Zhao et al., 2024) with a 20-step denoising process.

**Evaluation Metrics.** We evaluate the effectiveness of the attack algorithms from two perspectives: the performance of watermark attacks and the visual quality of the attacked images. Watermark robustness is measured by the Bit Error Rate (BER), which calculates the percentage of incorrectly decoded bits. A BER of 0% indicates perfect decoding, while 50% represents a complete failure, equivalent to random guessing.

In order to evaluate the visual quality of the attacked images, we employ a range of reference and no-reference metrics. For reference metrics, we employ PSNR and SSIM (Wang et al., 2004) to quantify the pixel difference between the attacked image and its original image. For no-reference metrics, we use PIQE (Venkatanath et al., 2015) and BRISQUE (Mittal et al., 2012), HYPERIQA (Su et al., 2020), CLIPIQA (Wang et al., 2023), and MUSIQ (Ke et al., 2021) to measure the visual

quality of the attacked image itself. A detailed introduction to each specific metric is provided in Appendix D.

**SRAttack Instantiations.** To implement our SRAttack framework, we instantiate the key components as follows. For the pre-degradation ($\mathcal{A}_{\text{pre}}$), we employ the Resize operation. Specifically, we employed bicubic interpolation to downscale the watermarked image to 25% of its original dimensions. We instantiate two super-resolution models to demonstrate the efficacy of our framework: **Real-ESRGAN** (Wang et al., 2021b) and **AdcSR** (Chen et al., 2025), and the default scaling factor is 4. In the subsequent experiments, we denote the SRAttack instantiated by Real-ESRGAN as SRAttack-1, and the SRAttack instantiated by AdcSR as SRAttack-2. We also employ bicubic interpolation as the final downscaling stage.

Table 1: Watermark bit error rate (BER, %) under various attacks. We highlight values greater than 45% in **bold**, as they signify that the watermark is almost completely destroyed.

| Attack Type | DwtDct | DwtDctSvd | RivaGAN | StegaStamp | IWRN | CIN | Avg. |
|---|---|---|---|---|---|---|---|
| No Attack | 17.33 | 2.27 | 1.05 | 0.01 | 0.00 | 0.01 | 3.45 |
| Gaussian Blur | 40.33 | 2.58 | 1.19 | 0.01 | 0.00 | 2.80 | 7.82 |
| Gaussian Noise | 26.39 | 3.35 | 2.24 | 0.10 | 0.00 | 3.73 | 5.97 |
| Contrast | **51.56** | **50.54** | 35.44 | 0.15 | 2.59 | 0.01 | 23.39 |
| Brightness | **51.54** | **50.53** | 38.31 | 0.16 | 0.00 | 0.01 | 23.43 |
| Rotate | **47.83** | 43.41 | 10.60 | 16.35 | 14.33 | 50.46 | 22.69 |
| Resize | **47.07** | 3.77 | 6.64 | 0.04 | 24.38 | 14.83 | 16.13 |
| JPEG | **49.54** | 23.43 | 10.81 | 0.88 | 43.01 | 30.02 | 26.29 |
| BM3D Denoise | **50.18** | 38.01 | 27.58 | 1.27 | **49.54** | 21.46 | 31.34 |
| VAE-Bmshj | **50.03** | 23.20 | 28.02 | 0.97 | **48.38** | 34.90 | 30.92 |
| VAE-Cheng | **49.96** | 25.96 | 25.63 | 1.48 | 28.73 | 27.34 | 26.52 |
| Diffusion | **50.21** | 36.74 | 38.53 | 6.45 | 42.50 | 33.24 | 34.62 |
| SRAttack-1 | **50.15** | **45.58** | 44.08 | 22.01 | 43.71 | **49.17** | 42.45 |
| SRAttack-2 | **50.17** | **48.40** | **48.61** | 36.71 | **50.92** | **48.50** | **47.22** |

## 4.2 RESULTS AND ANALYSIS

### 4.2.1 EFFECTIVENESS OF ATTACKS (RQ1)

To answer **RQ1**, we evaluate the effectiveness of SRAttack by measuring the Bit Error Rate (BER) of the extracted watermarks. As shown in Table 1, conventional image processing attacks only demonstrate effectiveness against traditional watermarking methods. Learning-based compression attacks outperform image processing-based attacks, particularly for INN-based and Encoder-Decoder watermarking schemes. Diffusion attacks show comparable performance on both traditional and INN-based watermarks, but achieve superior results against Encoder-Decoder architectures.

The proposed SRAttack-1 and SRAttack-2 exhibit consistently higher average bit error rates than competing methods. Evaluated under six threat models, they achieve BERs of 42.45% and 47.22% respectively, representing significant improvements over current watermark attack techniques. For traditional watermarking threat model, both SRAttack-1 and SRAttack-2 achieve BER values exceeding 45%, indicating near-complete watermark destruction. When attacking Encoder-Decoder based watermarking threat models, our framework still deliver optimal BER performance. Against INN-based watermarking approaches, the attacks consistently produce BER values above 43%. These results demonstrate exceptionally effective attack performance.

### 4.2.2 VISUAL QUALITY OF ATTACKED IMAGES (RQ2)

We evaluated the visual quality of the images after the attack to answer **RQ2**. The results, measured by a diverse set of seven metrics, are presented in Table 2.

Table 2: Visual quality evaluation of attacked images, using StegaStamp as the victim. Arrows indicate preferred values (↑ meaning higher is better, ↓ meaning lower is better).

| Attack / State | PIQE ↓ | BRISQUE ↓ | PSNR ↑ | SSIM ↑ | HyperIQA ↑ | CLIPIQA ↑ | MUSIQ ↑ |
|---|---|---|---|---|---|---|---|
| Cover Image | 34.88 | 25.59 | ∞ | 1.00 | 0.51 | 0.61 | 48.20 |
| Watermarked Image | 28.69 | 23.29 | 34.23 | 0.97 | 0.47 | 0.52 | 46.74 |
| Gaussian Blur | 53.32 | 41.13 | 27.56 | 0.83 | 0.29 | 0.37 | 29.47 |
| Gaussian Noise | 49.68 | 41.95 | 25.46 | 0.58 | 0.39 | 0.44 | 40.06 |
| Contrast | 40.87 | 37.26 | 14.64 | 0.51 | 0.47 | 0.35 | 39.58 |
| Brightness | 38.94 | 37.52 | 6.60 | 0.11 | 0.42 | 0.40 | 26.77 |
| Rotate | 26.52 | 26.99 | 12.09 | 0.28 | 0.49 | 0.38 | 44.04 |
| Resize | 79.28 | 53.67 | 22.79 | 0.66 | 0.21 | 0.44 | 22.25 |
| JPEG | 37.56 | 32.20 | 29.61 | 0.87 | 0.40 | 0.51 | 43.18 |
| BM3D Denoise | 91.64 | 69.74 | 23.79 | 0.61 | 0.26 | 0.28 | 23.46 |
| VAE-Bmshj | 52.96 | 34.12 | 29.69 | 0.87 | 0.47 | 0.43 | 47.45 |
| VAE-Cheng | 56.73 | 34.46 | **31.11** | **0.90** | 0.51 | 0.48 | 50.45 |
| Diffusion | 26.46 | 20.78 | 23.60 | 0.64 | 0.45 | 0.51 | 43.89 |
| SRAttack-1 | 28.14 | 22.45 | 21.54 | 0.60 | 0.54 | 0.46 | 45.28 |
| SRAttack-2 | **24.06** | **16.20** | 20.38 | 0.54 | **0.64** | **0.69** | **55.65** |

Experimental results demonstrate that SRAttack-2 achieves optimal performance across all five no-reference quality assessment metrics (specifically designed to predict human visual perception), significantly outperforming other attack methods while achieving comparable or even superior results to the original "cover images" and "watermarked images." SRAttack-1 also delivers outstanding visual quality compared to other attack methods. Although SRAttack does not achieve the highest PSNR and SSIM scores, this is expected since these metrics measure pixel-wise similarity to the original image—higher values indicate closer resemblance to the source. As an image regeneration method, SRAttack's primary objective is to generate new images with enhanced visual quality for watermark removal rather than precisely reconstructing the original pixels. Due to the superior perceptual quality of the generated images, they exhibit greater pixel-level deviations from the original. Detailed visual comparisons are provided in Appendix E.

Table 3: Impact of pre-degradation ($\mathcal{A}_{pre}$). SRA denotes the final two steps of SRAttack. '+ SRA' indicates that the 'Base' method serves as the pre-degradation step. The victim model is StegaStamp.

| Base Attack | BER (%) ↑ | | BRISQUE ↓ | | CLIPIQA ↑ | |
|---|---|---|---|---|---|---|
| | Base | + SRA | Base | + SRA | Base | + SRA |
| No Attack | 0.01 | 2.15 | 23.29 | 28.83 | 0.52 | 0.71 |
| Gaussian Blur | 0.01 | 9.38 | 41.13 | 26.59 | 0.37 | 0.69 |
| Gaussian Noise | 0.10 | 15.78 | 41.95 | 34.95 | 0.44 | 0.70 |
| VAE-Bmshj | 0.97 | 4.17 | 34.12 | 29.95 | 0.43 | 0.63 |
| VAE-cheng | 1.48 | 5.19 | 34.46 | 32.96 | 0.48 | 0.69 |
| BM3D Denoise | 1.27 | 36.43 | 69.74 | 46.41 | 0.28 | 0.45 |
| JPEG | 0.88 | 38.54 | 32.20 | 32.11 | 0.51 | 0.48 |
| Resize | 0.04 | 22.01 | 53.67 | 22.45 | 0.44 | 0.46 |

### 4.2.3 Impact of Pre-degradation on SRAttack (RQ3)

To answer **RQ3**, we conducted an ablation study and selected several straightforward and common attack methods as pre-degradation operations. The results shown in Table 3. When SRAttack is applied without any pre-degradation ("SRAttack w/o $\mathcal{A}_{pre}$"), it is largely ineffective, achieving a Bit Error Rate (BER) of only 2.15%. In contrast, applying SRAttack after an initial degradation attack dramatically increases its effectiveness. The results indicate that JPEG compression as a pre-degradation step yields the highest BER 38.54%. The Resize operation presents a balanced approach. While its BER of 22.01% is highly effective at destroying the watermark. Crucially, it

achieves this with minimal impact on perceptual quality, yielding the best BRISQUE score (22.45). This score is remarkably close to the "No Attack" baseline of 23.29, indicating that the resulting image quality is very well preserved. In summary, the pre-degradation step is an essential component of SRAttack. Jpeg providing the most powerful attack and Resize ensuring a balance between watermark destruction and image quality.

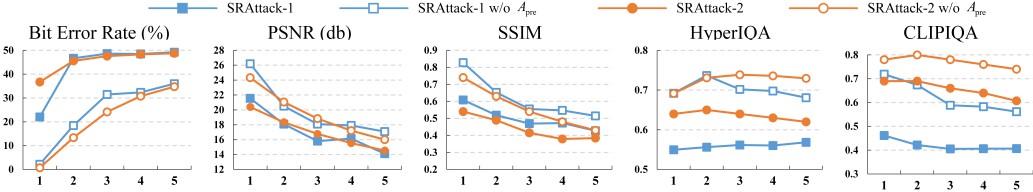

Figure 3: Effect of applying SRAttack multiple times. The horizontal axis indicates the number of repeated attacks, for example, 4 indicates four applied repeated attacks.

### 4.2.4 IMPACT OF HYPERPARAMETERS (RQ4)

The key hyperparameters in our framework primarily include the upsampling factor of the super-resolution model and the number of iterative attack times. We evaluated the performance of SRAttack under varying configurations of these hyperparameters.

**Effect of Iterative Attacks.** We investigate the effects of repeatedly applying SRAttack on the same image similar to WAVES (Rabbani et al., 2024). We conduct up to five rounds of attacks with SRAttack-1, SRAttack-2, and their variants without pre-degradation. The results are presented in Figure 3. From the Bit Error Rate (BER) plot, we observe that iterative attacks significantly enhance watermark removal effectiveness. For 'SRAttack-1' and 'SRAttack-2', the BER apidly increases and approaches the 50% threshold within just two iterations. Their "w/o $\mathcal{A}_{\text{pre}}$" versions start with a very low BER but show a substantial and steady increase with each iteration. This demonstrates that increasing the number of attack times can progressively enhance the attack effectiveness, but the improvement tends to saturate when reaching a certain iteration threshold. All methods exhibit progressively decreasing PSNR and SSIM values with increasing iterations, demonstrating growing divergence from the original images. The HyperIQA score initially increases and then decreases with the number of attack iterations, while the CLIPIQA score progressively decreases before stabilizing. Overall, the selection of iteration numbers requires careful consideration to balance between attack efficacy and visual quality.

**Upscaling Factor.** We investigate the impact of different upscaling factors (2× and 4×) using the Real-ESRGAN model, finding that larger factors achieve more effective watermark removal at the cost of visual quality, while smaller factors better preserve quality but are less effective at removing watermarks. The amplification factor can be regarded as an adjustable parameter for balancing attack effectiveness and visual fidelity. Detailed results are provided in Appendix F.

## 5 CONCLUSION

In this paper, we proposed SRAttack, a framework that leverages super-resolution (SR) models to effectively remove invisible watermarks while maintaining or even enhancing image quality. The proposed framework contains three components: Pre-degradation, Super-Resolution Reconstruction, and Downscaling. These three steps are effectively integrated. We conducted various experiments, the results confirm that SRAttack outperforms existing methods, achieving 47.22% average BER and the best visual quality in five no-reference metrics. We also conducted ablation studies and hyperparameter experiments to validate the effectiveness of the proposed method. Although we have implemented and validated two instantiation methods in our experiments, future research could further explore alternative instantiations of SRAttack or investigate other approaches to enhance image quality in adversarial attacks.

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

## A  ATTACK MECHANISM ANALYSIS

To better understand the underlying attack mechanisms of different methods, we provide a detailed analysis of their effects in the frequency domain. By examining the power spectrum of the attacked images, we can visualize how each method interacts with spatial frequency components and thereby infer how watermark signals are disrupted or removed. The diagram in Figure 4.

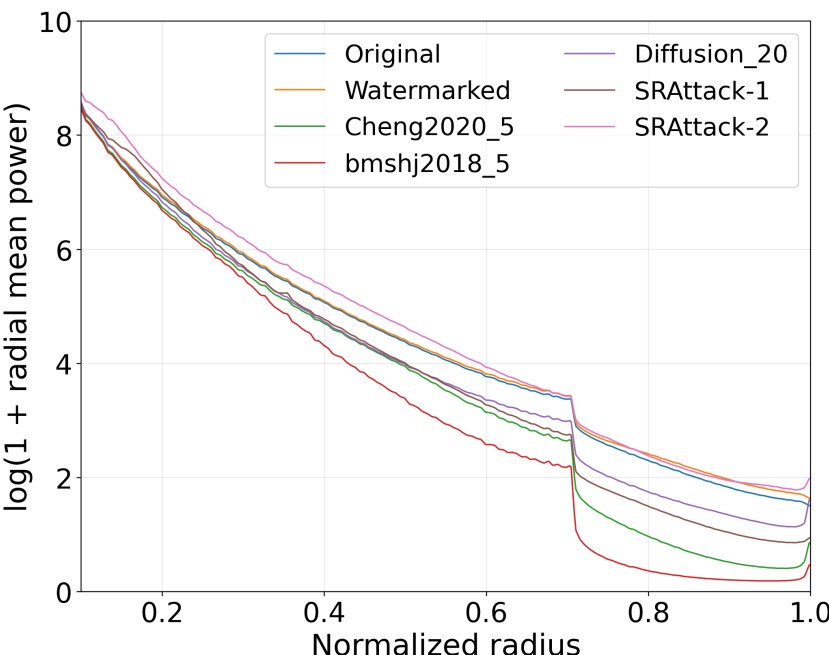

Figure 4: The mean radial average power spectrum of the attacked images. The x-axis represents the spatial frequency, while the y-axis shows the power of the frequency components of the image.

The watermarked image remains close to the original in terms of overall distribution. We have observed that compression algorithms significantly reduce the image power spectrum, while reconstruction-based methods can better preserve the power distribution. SRAttack-1 (AdcSR) tends to enrich details in the mid-frequency range, generating synthetic textures that enhance perceptual realism and confuse the watermark decoder. Additionally, in the high-frequency range, the curve is closer to the original and watermarked curves, but with a distinct trajectory. This suggests that SRAttack-1 actively reconstructs high-frequency details. In contrast, SRAttack-2 (Real-ESRGAN) and Diffusion show an overall reduction across the mid- and high-frequency bands, smoothing the image and removing artifacts to produce a clean, high-definition appearance. This indicates that SRAttack-1 disrupts watermark decoding primarily through detail regeneration, while SRAttack-2 and Diffusion achieve watermark removal through global spectral suppression.

Table 4: A list of representative Super-Resolution models and tools.

| Model/Tool | Key Feature | Avail./Ref. |
|---|---|---|
| **SRGAN** | First to introduce GANs to SR, generating images with better perceptual quality and realistic textures. | (Ledig et al., 2017) |
| **SwinIR** | Achieves strong performance based on the Swin Transformer architecture. | (Liang et al., 2021) |
| **BasicSR** | An open-source toolbox integrating a wide range of mainstream SR models and training/testing codes. | (Wang et al., 2022b) |
| **SeeSR** | A state-of-the-art image super-resolution model based on Stable Diffusion. | (Wu et al., 2024b) |
| **GFPGAN** | A practical algorithm for real-world face restoration. | (Wang et al., 2021a) |
| **Final2x** | A powerful tool that can use super-resolution models including Real-ESRGAN, SwinIR, HAT, SRCNN, and more. | GitHub |
| **DeepAI** | An online website that allows for quick and free use of super-resolution features to enhance images without login. | Online |

## B    LIST OF METHODS AND TOOLS FOR SRATTACK INSTANTIATION

This appendix provides a list of models and tools that can be used to instantiate the Super-Resolution (SR) operator within the SRAttack framework. It should be noted that our attack method can be instantiated not only with the SR models listed, but also with almost any SR model. The SR operator can be any off-the-shelf, pre-trained super-resolution model. The following table presents a selection of representative models and open-source tools, covering a range from classic to state-of-the-art methods, as shown in Table 4.

## C    ADDITIONAL EXPERIMENTAL DETAILS

### C.1    TRAINING CONFIGURATION

The threat models requiring training are StegaStamp, IWRN, and CIN. We trained these models from scratch using a unified setting to ensure a fair comparison.

**Training Dataset.**    We constructed a dedicated training set by randomly sampling 5,000 images from the MS-COCO (Lin et al., 2014) training set, 5,000 images from the WikiArt dataset, and 500 high-resolution images from the DIV2K (Timofte et al., 2017) dataset. This selection was made to be disjoint from our test set, ensuring no data overlap. For training, each image was processed by randomly cropping a patch and resizing it to $256 \times 256$ pixels.

**Training Strategy.**    All models were trained to embed a 32-bit random watermark. The training process was divided into two phases. First, we trained each model for 5 epochs without any distortions to establish a baseline for imperceptibility and message embedding. Following this, we trained for an additional 10 epochs with a set of adversarial distortions applied randomly to the watermarked images. This distortion set included: 'Identity', 'Resize', 'Gaussian Blur', 'Jpeg', 'Gaussian Noise', 'Brightness', 'Contrast', and 'Rotation'. This two-phase strategy helps the models learn robust representations. For IWRN, we disabled the SimSiam module during training, as its inclusion led to instability in our experiments. For all other aspects, we kept the training parameters consistent with the original implementations of each respective paper.

### C.2    WATERMARKING ALGORITHM DETAILS

This section provides a detailed description of the watermarking algorithms evaluated in our research, including their basic principles and implementation sources.

**DwtDct and DwtDctSvd** (Navas et al., 2008) are classic algorithms that operate in the frequency domain. They utilize a cascaded Discrete Wavelet Transform (DWT) and Discrete Cosine Transform (DCT) to embed a watermark, with the latter method further incorporating Singular Value Decomposition (SVD) for enhanced robustness by modifying the more stable singular values. This method has been used to watermark stable diffusion models (Rombach et al., 2022). We use the implementation from `https://github.com/ShieldMnt/invisible-watermark`.

**RivaGAN** (Zhang et al., 2019) a GAN-based framework. Its core mechanism is a dual-adversary system that jointly optimizes for imperceptibility (via a critic network) and robustness (via an attack network that simulates watermark removal). We use the implementation from `https://github.com/ShieldMnt/invisible-watermark`.

**StegaStamp** (Tancik et al., 2020) is a highly robust CNN-based watermarking model. Its key innovation lies in its training method, in which various image distortions are simulated and directly incorporated into the training loop as a form of adversarial attack. We have re-implemented StegaStamp based on PyTorch.

**IWRN** (Kou et al., 2025) is a framework designed around invertible neural networks (INN). The reversible architecture enables perfect and lossless reconstruction of the original image from the watermarked version, thereby ensuring high imperceptibility. To enhance robustness, IWRN employs a deformable attention mechanism and a learnable wavelet network that adaptively embeds the watermark into the imperceptible high-frequency regions of the image, where it is more resilient to attacks. We conduct experiments using the official code.

**CIN** (Ma et al., 2022) is a hybrid framework that combines reversible and irreversible networks. Its reversible network is used to achieve lossless embedding under normal conditions. In the case of strong noise such as JPEG, it first identifies whether the attack type is strong noise, and then uses the irreversible network to enhance the reversible network. We conduct experiments using the official publicly available code.

### C.3 IMPLEMENTATION DETAILS

All experiments were run on a server equipped with two Intel® Xeon® Gold 6530 CPUs, 256 GB of system RAM, and a single NVIDIA GeForce RTX 4090D GPU with 24 GB of VRAM. The software environment is built on Ubuntu 22.04 LTS, using Python 3.9.20. Our implementation is based on the PyTorch 2.1.0 framework, accelerated by CUDA 11.8 and cuDNN 8.7.0. Default conduct once attack.

## D  IMAGE QUALITY METRICS

This section provides a detailed overview of the no-reference image quality assessment (NR-IQA) metrics used in our study. These are categorized into traditional statistical methods (PIQE, BRISQUE) and modern learning-based approaches (HYPERIQA, CLIPIQA, MUSIQ).

**PIQE (Perceptual Image Quality Evaluator)** PIQE (Venkatanath et al., 2015) is an **unsupervised** and training-free no-reference image quality assessment model. It computes local distortion metrics by dividing the input image into non-overlapping blocks. Specifically, PIQE identifies high-activity regions (containing strong edges and textures) and low-activity, spatially smooth regions to estimate the presence of common distortions such as blocking artifacts and noise. These local distortion metrics are then aggregated into a global quality score. For this metric, a lower score indicates better visual quality.

**BRISQUE (Blind/Referenceless Image Spatial Quality Evaluator)** BRISQUE (Mittal et al., 2012) leverages the principles of Natural Scene Statistics (NSS) for **supervised** no-reference quality assessment. It posits that visual distortions alter the statistical properties found in pristine images. To quantify this, BRISQUE extracts features from Mean Subtracted Contrast Normalized (MSCN) coefficients that capture deviations from natural scene statistics. A regression model then maps these features to a perceptual quality score. For this metric, a lower score indicates better visual quality.

**HYPERIQA (Hyper-Network-based Image Quality Assessment)** HYPERIQA (Su et al., 2020) is a **content-adaptive** deep learning-based NR-IQA model featuring a hyper-network architecture. This design enables it to adapt its quality assessment to specific image content, leading to robust performance. Higher scores indicate better visual quality. For this metric, a higher score indicates better visual quality.

**CLIPIQA (Contrastive Language-Image Pre-training for IQA)** CLIPIQA (Wang et al., 2023) formulates No-Reference IQA as a problem of semantic similarity, building upon the pre-trained **vision-language model** CLIP (Radford et al., 2021). The model infers quality by computing the semantic alignment between an image's embedding and the embeddings of textual prompts representing good and poor quality. This approach leverages the rich semantic space of the multi-modal model, with higher similarity to the "good quality" prompt corresponding to a higher quality score. For this metric, a higher score indicates better visual quality.

**MUSIQ (Multi-scale Image Quality Transformer)** MUSIQ (Ke et al., 2021) model leverages a Vision Transformer (ViT) architecture for no-reference quality assessment. Its key innovation is a **multi-scale** framework that feeds image patches from varying resolutions into the Transformer encoder. This strategy enables the model to effectively capture long-range dependencies and simultaneously assess local and global visual characteristics, leading to a more holistic quality prediction. For this metric, a higher score indicates better visual quality.

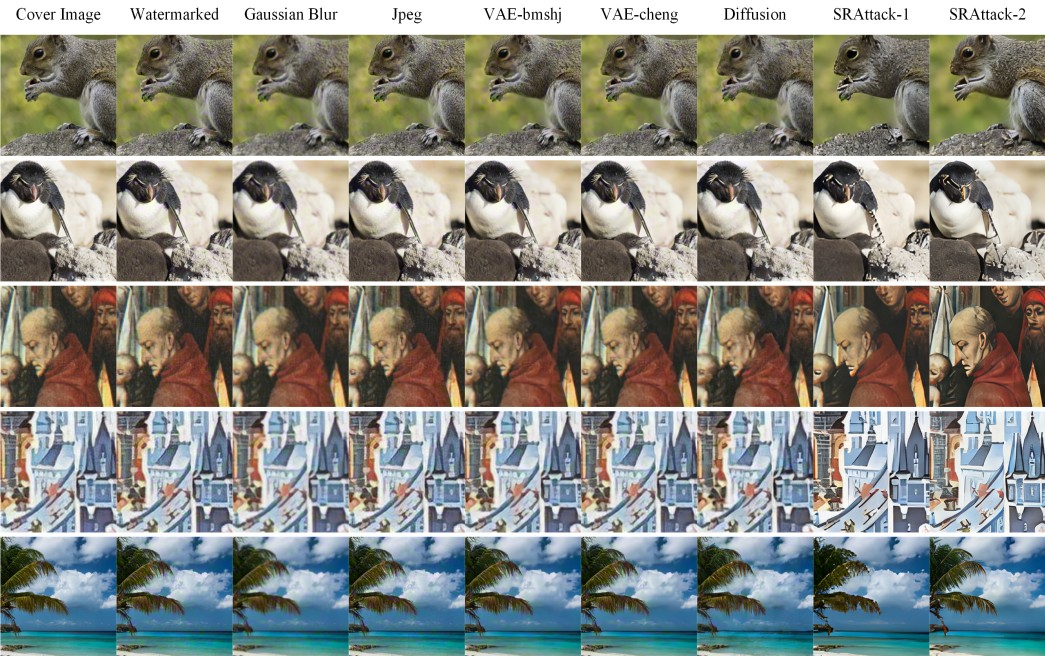

Figure 5: Visual comparison of different watermark attacks on images from the StegaStamp. Showing the cover image, watermarked image, and the outputs of various attack methods.

# E   VISUAL COMPARISON

We provide a qualitative comparison to visually demonstrate the effectiveness of our proposed SRAttack. Figure 5 presents the performance across diverse images. The results show that the image clarity of SRAttack-1 and SRAttack-2 is relatively good. When the quality of the original image is low, our method can significantly improve its clarity. Compared with SRAttack-1 and other attacks, SRAttack-2 provides better texture detail supplementation. For artifacts introduced by watermarking algorithms, other attacks tend to preserve them rather than eliminate them.

## F    IMPACT OF UPSCALING FACTOR

We provide a qualitative comparison to visually demonstrate the effectiveness of our proposed SRAttack. Figure 6 presents the performance across diverse images. The results show that the image clarity of SRAttack-1 and SRAttack-2 is relatively good. When the quality of the original image is low, our method can significantly improve its clarity. Compared with SRAttack-1 and other attacks, SRAttack-2 provides better texture detail supplementation. For artifacts introduced by watermarking algorithms, other attacks tend to preserve them rather than eliminate them.

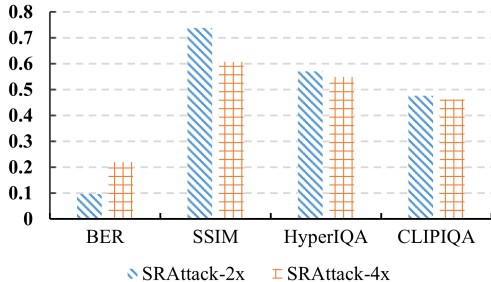

Figure 6: Impact of the SR scaling factor on SRAttack's performance.

## G    THE USE OF LARGE LANGUAGE MODELS

This section is written in accordance with the requirements of the ICLR author guidelines regarding the use of Large Language Models (LLMs) in the paper. The main LLMs used in this paper's writing is Gemini-2.5-Pro. The LLMs was utilized for translation, grammar checking of the writing, and partial sentence polishing. Additionally, the LLMs assisted in the transcription of the paper's LaTeX code, such as adjusting the LaTeX format.

