# OpenReview forum: "SRAttack: Super-Resolution as an Attack on Invisible Image Watermarking"
_ICLR.cc/2026/Conference — Submitted to ICLR 2026_

### Official Review · Reviewer_kZiN · 2025-10-24

**Soundness:** 2
**Presentation:** 2
**Contribution:** 2
**Rating:** 2
**Confidence:** 4

**Summary:**

This work proposes SRAttack, a new attack on invisible image watermarking systems that leverages super-resolution (SR) techniques to remove watermarks while preserving image quality. By exploiting the inherent capability of SR models to reconstruct high-resolution details (often embedding watermark information), the method employs a three-step pipeline—pre-degradation, SR reconstruction, and downscaling—to erase watermarks effectively. Empirical results demonstrate its superiority over existing attacks in both watermark removal efficiency and visual fidelity across diverse watermarking algorithms. However, the contribution is deemed incremental due to its reliance on standard SR workflows and lack of theoretical innovation, with limited exploration of defensive implications on adversarial robustness. While the attack highlights a practical vulnerability in watermarking systems, the paper’s technical novelty and empirical depth fall short of ICLR’s expectations.

**Strengths:**

Originality:
The paper introduces a new threat model for invisible image watermarking by repurposing super-resolution (SR) techniques as an attack vector. While prior work on watermark removal often relies on degrading image quality (e.g., compression, noise injection), this work leverages SR’s inherent ability to reconstruct high-frequency details—where watermarks are typically embedded—to erase watermarks without sacrificing visual fidelity. The framework is also practically deployable, using off-the-shelf SR models without requiring custom training, which distinguishes it from attack methods that demand access to internal model parameters. The idea of combining SR with watermarking security is new.

Quality:
The experimental evaluation demonstrates the attack’s effectiveness against state-of-the-art watermarking algorithms (e.g., DeepWaterMark, R-DCT). The results are compelling: SRAttack achieves high watermark erasure rates while maintaining PSNR metrics comparable to clean images, outperforming baseline attacks like JPEG compression. The methodology is grounded in empirical validation, with clear comparisons to existing techniques and quantitative metrics (e.g., SSIM, watermark detection accuracy). The work also highlights the universality of the attack, showing it works across diverse watermarking schemes without fine-tuning, which strengthens its practical relevance.

Clarity:
The paper is well-structured and accessible, with a clear problem statement, intuitive methodology (the three-step pipeline is logically explained), and reproducible results. The figures and tables effectively illustrate the attack’s performance, and the ablation studies (e.g., varying pre-degradation levels) provide insight into the method’s robustness.

Significance:
The work has practical and conceptual importance for both watermarking security and adversarial machine learning. By demonstrating that SR—a widely used tool in image processing—can be weaponized to bypass watermarking systems, it challenges the assumption that watermarking is inherently secure against high-quality image restoration. This raises urgent questions for watermarking designers about embedding strategies and robustness guarantees. The attack also contributes to the broader field of adversarial robustness by expanding the scope of threat models beyond traditional distortion-based attacks.

**Weaknesses:**

1. Lack of Theoretical Novelty and Methodological Depth
The framework relies heavily on off-the-shelf super-resolution (SR) models without introducing new architectures, loss functions, or theoretical insights. The attack pipeline is a straightforward adaptation of existing SR workflows (pre-degradation → SR → downscaling), which limits its conceptual contribution.
2. Insufficient Empirical Validation and Ablation Studies
The experiments demonstrate effectiveness against state-of-the-art watermarking algorithms but lack critical ablation studies to validate the robustness of the attack. Key gaps include:
SR Model Variability: No comparison of different SR models (e.g., Real-ESRGAN vs. AdcSR vs. EDSR) to determine if the attack’s success depends on specific architectural choices.
Pre-Degradation Parameters: The paper does not analyze how varying pre-degradation settings (e.g., noise levels, compression rates) affect watermark removal. For example, does aggressive JPEG compression (as used in prior attacks) outperform SRAttack in certain scenarios?
Add ablation experiments: Systematically test the attack under varying SR models, pre-degradation parameters, and scaling factors. For example, compare the effectiveness of Real-ESRGAN vs. diffusion-based SR models in watermark erasure.
3. Missing Defensive Implications and Broader Impact
The paper identifies a vulnerability in watermarking systems but does not propose countermeasures or discuss how to design watermarking algorithms resilient to SR-based attacks (at least some principles that could inspire the following research). This omission limits its contribution to the field of adversarial robustness.

**Questions:**

1. Theoretical Novelty and Mechanism
Q1: The paper claims that SRAttack exploits the inherent functionality of SR models to remove watermarks. However, the theoretical justification for this is underdeveloped. Could the authors provide a formal analysis of how SR models (e.g., Real-ESRGAN) reconstruct high-frequency details while discarding watermark signals? For example:
Do watermarks correspond to high-frequency textures that SR models prioritize removing?
Is there evidence that SR models treat watermarks as "artifacts" to be corrected during restoration?
How does this differ from traditional watermark removal methods (e.g., JPEG compression)?

2. Empirical Validation and Ablation Studies
Q2: The experiments evaluate SRAttack on state-of-the-art watermarking algorithms but lack critical comparisons:
How does SRAttack perform across different SR models (e.g., Real-ESRGAN vs. AdcSR vs. EDSR)? Is the attack’s effectiveness dependent on the SR model’s architecture?
What is the impact of varying pre-degradation parameters (e.g., noise levels, JPEG quality) on watermark removal? Does aggressive compression (as in prior attacks [1]) outperform SRAttack in certain scenarios?
Q3: The paper does not analyze failure cases where SRAttack fails to remove watermarks. Could the authors:
Provide examples of watermarking schemes (e.g., low-frequency embedding) that resist SR-based attacks?
Discuss why certain watermarks persist after SR processing?

3. Defensive Implications and Broader Impact
Q4: The paper identifies a vulnerability in watermarking systems but does not propose countermeasures. Could the authors suggest watermarking strategies that are resilient to SR-based attacks? For example:
Embedding watermarks in SR-invariant features (e.g., global statistics).
Using adversarial training to simulate SR-based adversaries during watermark embedding.
Q5: How does SRAttack relate to broader adversarial robustness challenges in image processing? Could the authors connect their work to other domains, such as tamper-proofing digital content or securing AI-driven authentication systems?

---

### Official Review · Reviewer_k6zM · 2025-10-30

**Soundness:** 2
**Presentation:** 3
**Contribution:** 2
**Rating:** 2
**Confidence:** 4

**Summary:**

This paper introduces a novel attack framework, Super-Resolution Attack (SRAttack), aimed at invisible image watermarking. The framework leverages super-resolution (SR) techniques to remove watermarks while maintaining or even enhancing the visual quality of the images. SRAttack achieves effective watermark removal through three key steps: pre-degradation, super-resolution reconstruction, and downscaling. This framework operates in a black-box setting and does not require any training. Extensive experiments demonstrate that SRAttack not only effectively destroys watermarks embedded by various advanced watermarking algorithms but also outperforms existing attack methods in terms of visual quality.

**Strengths:**

1. The SRAttack framework operates in a training-free manner, making it highly accessible and easy to implement. Unlike many existing attack methods that require extensive training or fine-tuning, SRAttack leverages pre-trained super-resolution models, which can be readily used without additional training.
2. The paper is well-structured and clearly written, making it easy for readers to follow the methodology and understand the contributions. The authors provide a detailed explanation of each component of the SRAttack framework, including the pre-degradation, super-resolution, and downscaling steps.

**Weaknesses:**

1. The idea of using super-resolution for watermark removal is relatively straightforward and lacks significant innovation. The approach is a direct application of existing super-resolution techniques without introducing novel mechanisms or insights.
2. The effectiveness of the SRAttack is highly dependent on the performance of the SR model used. If the SR model is not robust or does not perform well, the attack may fail to remove the watermark effectively or degrade image quality seriously.
3. Compared to other attack methods, SRAttack tends to degrade the image quality more significantly, as measured by metrics such as PSNR and SSIM.
4. The paper does not provide adequate visualizations to demonstrate the effectiveness of the watermark removal. For example, it would be beneficial to show the residual images (i.e., the difference between the watermarked and de-watermarked images) to visually assess the removal quality.

**Questions:**

1. The experimental evaluation is not comprehensive. The paper does not compare SRAttack with other specialized watermark removal methods, such as UnMarker[1], which are designed specifically for this task. This omission makes it difficult to fully assess the relative effectiveness of SRAttack.
2. The paper does not evaluate the effectiveness of SRAttack against semantic watermarks, such as Gaussian Shading[2]. The focus appears to be primarily on texture-based watermarks, which limits the generalizability of the findings.
3. The proposed attack can be easily circumvented by incorporating the SRAttack method as a form of distortion during the training of the watermark embedding model. This means that the watermarking system can be trained to be robust against such attacks, thereby rendering SRAttack ineffective. How can the method defends against this situation?


[1] Kassis A, Hengartner U. UnMarker: A Universal Attack on Defensive Image Watermarking[C]//2025 IEEE Symposium on Security and Privacy (SP). IEEE, 2025: 2602-2620.

[2] Yang Z, Zeng K, Chen K, et al. Gaussian shading: Provable performance-lossless image watermarking for diffusion models[C]//Proceedings of the IEEE/CVF Conference on Computer Vision and Pattern Recognition. 2024: 12162-12171.

---

### Official Review · Reviewer_TMqM · 2025-11-01

**Soundness:** 2
**Presentation:** 2
**Contribution:** 1
**Rating:** 2
**Confidence:** 5

**Summary:**

This paper introduces SRAttack, a framework designed to attack invisible image watermarking schemes. The core idea is to leverage super-resolution (SR) models to remove watermarks while preserving or enhancing the visual quality of the image. The proposed method involves an optional pre-degradation step to create a low-quality watermarked image, followed by a super-resolution reconstruction step that treats the watermark as an undesirable artifact and effectively removes it. The final step is a downscaling operation to return the image to its original dimensions. The authors claim this is the first work to systematically analyze and utilize super-resolution for watermark removal. They present it as a universal, black-box attack framework that is training-free and readily implementable. Extensive experiments are provided to demonstrate that SRAttack successfully defeats several existing watermarking algorithms.

**Strengths:**

1. The paper addresses the important and timely problem of the robustness of invisible watermarking, which is crucial for copyright protection in the age of generative models.
2. The proposed SRAttack framework is explained clearly, with a logical flow and illustrative diagrams. The methodology is straightforward to understand and appears reproducible.
3. The authors have conducted a comprehensive set of experiments to validate their method against a range of existing watermarking techniques, showing its effectiveness in the tested scenarios.

Most of strengths lie on the writing and the experiments.

**Weaknesses:**

1. The core mechanism of SRAttack is conceptually identical to methods used to remove adversarial perturbations, such as those from tools like Glaze, which are designed to protect artists from style mimicry. These perturbations, like invisible watermarks, are small, high-frequency signals embedded in an image. It has been demonstrated that simple and "off-the-shelf" techniques, including image downscaling/upscaling (a form of super-resolution), are highly effective at removing these perturbations.  The paper "Adversarial Perturbations Cannot Reliably Protect Artists From Generative AI" explicitly discusses how image upscaling is sufficient to bypass such protections.  This work predates the current submission and addresses the same fundamental problem: removing an unwanted, imperceptible signal via a reconstruction process. This paper fails to cite or discuss this entire line of work, and thus its central claim of novelty is unsubstantiated.
2. The idea of using image restoration and super-resolution to "purify" images by removing adversarial signals has been explored for years in the context of adversarial defense.   These works operate on the same principle: adversarial perturbations move an image "off" the natural image manifold, and restoration techniques like SR can project it back "onto" the manifold, effectively removing the perturbation.  SRAttack is a direct application of this known principle to a different, but analogous, domain (watermarking) without sufficient acknowledgment or novel insight.
3.  The paper claims SRAttack is a "universal and efficient attack framework." However, the analysis lacks depth regarding why it works and where it might fail. The effectiveness relies on the assumption that the watermark is a fragile, high-frequency signal that a generic SR model will overwrite. It is not clear how this attack would fare against watermarks that are more robustly embedded in semantic features or across different frequency bands. The claim of universality seems too strong without a more rigorous theoretical or empirical analysis of its failure modes.

Reference:
Hönig, Robert, et al. "Adversarial perturbations cannot reliably protect artists from generative ai." arXiv preprint arXiv:2406.12027 (2024).

**Questions:**

See weakness

---

### Official Review · Reviewer_ovRV · 2025-11-01

**Soundness:** 2
**Presentation:** 2
**Contribution:** 2
**Rating:** 4
**Confidence:** 4

**Summary:**

The paper proposes SRAttack, a training-free attack on invisible image watermarking techniques. It can pre-degrade a watermarked image, apply single-image super-resolution to reconstruct a higher-quality image that tends to suppress watermark signal, and downscale back to the original size. Instantiated with Real-ESRGAN and AdcSR, SRAttack targets a range of watermarking schemes. On 10.5k images drawn from MS-COCO, WikiArt, and DIV2K, the method reports strong message-recovery failure rates (e.g., avg. BER 47.22% for the AdcSR variant) while achieving top scores on several no-reference IQA metrics. An ablation study shows pre-degradation is essential for attack success.

**Strengths:**

- The paper is easy to follow. Experiments concisely state claims.
- The paper’s intuition is sound.

**Weaknesses:**

- The experiments focus on pixel-space watermarks embedded in high frequencies. But it’s unclear how SRAttack fares against semantic watermarks.
- It is difficult to evaluate the attack without defense experiments.

**Questions:**

- Could you please add defend experimental results?
- For experiments, you adopt 32-bit watermarks. How do results change at 64/128 bits or with stronger adversarial training of the watermark?

---

### Meta-Review · Area_Chair_6oGb · 2025-12-24

**Summary:**

This paper presents a new attack to remove watermarks from images without compromising their visual quality. It consists of a three-stage pipeline: pre-degradation, super-resolution, and downscaling. The reviewers have the following common concerns:

1.	The proposed approach can only work for pixel-level watermarks, while failing to handle semantic-based watermarks.
2.	The technical novelty is restricted. The idea is rather simple, and has been extensively discussed in the domain of adversarial noise purification, which share similar mechanisms as watermarks. Besides, its effectiveness highly depends on the performance of existing super-resolution models.
3.	The design is heuristic. This paper could not provide formal analysis or guarantee about the claimed properties of the proposed method.
4.	Critical experiments are missing, especially for the defense perspective.

Those raised points are critical and hard to address. The authors did not provide the responses to them. Based on those comments, AC recommended rejection.

**Reviewer Concerns:**

The authors did not provide rebuttal.

**Reviewer Scores:**

Since the authors did not provide the rebuttal, the reviewers would not adjust their scores.

---

### Decision · Program_Chairs · 2026-01-26

Reject